# Addressing Gaps for Health Systems Strengthening: A Public Perspective on Health Systems’ Response towards COVID-19

**DOI:** 10.3390/ijerph18179047

**Published:** 2021-08-27

**Authors:** Nur Zahirah Balqis-Ali, Weng Hong Fun, Munirah Ismail, Rui Jie Ng, Faeiz Syezri Adzmin Jaaffar, Lee Lan Low

**Affiliations:** 1Institute for Health Systems Research, National Institutes of Health, Ministry of Health Malaysia, Shah Alam 40170, Selangor, Malaysia; fun.wh@moh.gov.my (W.H.F.); low.ll@moh.gov.my (L.L.L.); 2Institute for Health Management, National Institutes of Health, Ministry of Health Malaysia, Shah Alam 40170, Selangor, Malaysia; munirahismail@moh.gov.my (M.I.); ngrj@moh.gov.my (R.J.N.); faeiz@moh.gov.my (F.S.A.J.)

**Keywords:** health systems strengthening, health systems building blocks, systems thinking, COVID-19, Malaysia, outbreak, epidemic

## Abstract

Strengthening the health systems through gaps identification is necessary to ensure sustainable improvements especially in facing a debilitating outbreak such as COVID-19. This study aims to explore public perspective on health systems’ response towards COVID-19, and to identify gaps for health systems strengthening by leveraging on WHO health systems’ building blocks. A qualitative study was conducted using open-ended questions survey among public followed by in-depth interviews with key informants. Opinions on Malaysia’s health systems response towards COVID-19 were gathered. Data were exported to NVIVO version 12 and analysed using content analysis approach. The study identified various issues on health systems’ response towards COVID-19, which were then mapped into health systems’ building blocks. The study showed the gaps were embedded among complex interactions between the health systems building blocks. The leadership and governance building block had cross-cutting effects, and all building blocks influenced service deliveries. Understanding the complexities in fostering whole-systems strengthening through a holistic measure in facing an outbreak was paramount. Applying systems thinking in addressing gaps could help addressing the complexity at a macro level, including consideration of how an action implicates other building blocks and approaching the governance effort in a more adaptive manner to develop resilient systems.

## 1. Introduction

Health systems strengthening (HSS) involves a continuous process of implementing changes in policy and practice to improve access, coverage, quality and efficiency of the health system [1]. It is context-dependent and progressive, requiring regular assessments of the systems’ capacities and weaknesses, followed by regular review of the systems’ performance [2]. Essentially, HSS is about generating long-term changes, rather than filling identified gaps or implementing interventions with short-term benefits [3]. Elements of HSS therefore should be purposeful, developed in a rationale manner and sustainable [4,5]. However, the lack of consensus and unity on what defines the ‘standards’ of a strong or resilient health system pose a challenge in translating the theory into practice [6].

Multiple frameworks are available to guide countries in undertaking HSS activities and evaluation of HSS related programs [7]. The World Health Organisation (WHO) Health Systems Building Blocks is commonly used to describe the core components of a responsive and efficient health system and is widely adopted in various studies identifying elements needed to improve the health systems [8,9,10,11]. While the building blocks form the foundation of the health sector, the relationships and interactions among the building blocks represent the constituent of a health system. WHO declares HSS efforts must improve the interactions between the building blocks to achieve sustainable outcomes [1]. The design and implementation of HSS approaches are therefore fundamentally complex and require understanding of the dynamic interactions between the elements, organisations, actors and players, as well as adaptation to the constantly evolving context [11,12]. The practicality of using building blocks in analysing dynamic, complex and inter-linked health systems has been previously debated [13]. Nonetheless, this broad framework serves as a platform to describe the multifaceted response towards an event or crisis [8,14,15].

Strengthening the health systems traverses beyond aiming to achieve certain goals or pre-set outcomes, rather describes its ability to adapt and transform in responding effectively to an unanticipated crisis [8,16]. The unprecedented pandemic of novel coronavirus disease (COVID-19) caused by Severe Acute Respiratory Syndrome Coronavirus 2 (SARS-CoV-2) has affected almost all countries ever since the first case was recorded in Wuhan, China [17]. Without any definitive treatment, most health systems were overwhelmed and health services disrupted as the infection rapidly spread across the globe and the number of cases grew exponentially [18]. Past and recent outbreak experiences have demonstrated countries with strong and established health systems were able to adapt and produce good health outcomes in contrast with vulnerable health systems which struggled to respond effectively to adverse conditions. Countries like Taiwan and South Korea leveraged on their past experience of SARS and MERS outbreaks which exposed their nation’s limitations and prompted improvements to be made [19,20,21]. Countries rapidly formed new approach and guidelines in adapting to the challenges brought upon by COVID-19 [22,23,24]. These led to a call and opportunity to redesign the health systems and policy thinking in strengthening the health systems’ core elements in the light of a pandemic.

A country’s health systems structure and design greatly influence the outbreak management. The first case of COVID-19 in Malaysia was detected on 24 January 2020 with the first surge in cases was related to a religious congregation attended by more than 10,000 people from all over the country [25]. As of 16 January 2021, Malaysia recorded a total of 155,095 cases and 594 deaths [26]. In effort to keep the spread of COVID-19 cases controlled, the Movement Control Order (MCO) was introduced beginning on the 18 March 2020 [27], and reintroduced with targeted approach during the third wave of the outbreak in January 2021 [28]. With regards to the structure, as Malaysia has a dichotomous health system, comprising heavily subsidised public sector and fee-for-service private sector [29], the majority of COVID-19 cases management were concentrated in public facilities. A conjoined effort between both sectors is thus essential to optimise the country’s health resources in fighting the outbreak. Another crucial collaboration is engagement with subject matter experts for accurate data analysis, modelling and prediction. As the custodian of health-related data, the Ministry of Health Malaysia (MOH) underlines policy on data sharing within and beyond the ministry, entailing fulfilment of protocols to ensure confidentiality and data ownership [30]. Such an approach, however, may hinder timely and accurate data generation and analysis in facing an outbreak of such a magnitude. Without sufficient testing capacity, good surveillance systems and support from healthcare facilities, health systems might fail to respond effectively to the outbreak. For example, Malaysia used targeted testing to track all close contacts but were at risk of excluding other potential positive cases beyond the scope of the definition [31].

With such issues and intricacies, the need to identify areas to strengthen health systems is elemental. Capturing these areas through the lens of people directly affected by a casualty or have gone through a difficult time is invaluable. Evidence specified obtaining public’s opinion in decision-making led to improved service and quality [32]. By exploring the public perspective on health systems’ response towards COVID-19 in the early phase of the outbreak, we seek to identify gaps for HSS, leveraging on WHO building blocks. This contributes towards understanding the core areas for improvement during and beyond the outbreak, informing policymakers and key stakeholders in planning strategies for health systems’ strengthening in responding towards an outbreak, should another befall in the future.

## 2. Materials and Methods

This qualitative component was part of a larger mixed method study conducted to assess Malaysia health systems’ response to COVID-19. The qualitative data were captured using open-ended questions from online survey among public followed by in-depth interviews (IDI) with experts. The study was conducted in three stages as summarised in Figure 1.

### 2.1. Stage 1: Gathering Public’s Perspective to Identify Issues of Health Systems Response

We conducted an online survey with open-ended questions among public from March till April 2020 to identify issues related to Malaysia’s health systems response towards COVID-19.

#### 2.1.1. Study Tool

The survey tool was adopted and adapted from WHO COVID-19 Strategic Preparedness and Response Plan (SPRP) document, retained in the English language [33]. It consists of eight pillars from health systems’ preparedness and response areas, namely, (1) country-level coordination, planning, and monitoring; (2) risk communication and community engagement; (3) surveillance, rapid-response teams, and case investigation; (4) points of entry; (5) national laboratories; (6) infection prevention and control; (7) case management; and (8) operations support and logistics. The survey was pretested among few healthcare professionals with various backgrounds to obtain their feedback for further improvement on the flow and formatting of each question and response, sentence structure, feasibility of answering the questions including duration taken to complete the survey and platform of choice The revised version based on the feedback was used for the online survey.

#### 2.1.2. Study Recruitment and Data Collection

We aimed to gather variations of responses and recruited respondents from different backgrounds and experiences among general public, both working in health and non-health related areas, as well as public health specialists from universities and non-governmental bodies.

Information sheet was provided before starting the online survey and respondents may choose to consent to participate by clicking the consent button. Given the mixed and vast areas covered across pillars, respondents were allowed to select and answer one or more pillars most relevant to their background and expertise. All responses were captured using Qualtrics survey software. The survey link was sent to public through various platforms to assess their opinions and concerns on the management of COVID-19 by Malaysia. Few approaches were used to reach the respondents including relay of messages through professional/personal networks, and sharing through the organisation website, Twitter, Facebook, WhatsApp and email. We also encouraged them to share the link among their network using snowball sampling approach. Few of the respondents identified as experts based on their diverse background and experience in health systems and community outreach crisis (public health specialists, clinical specialists, health committees’ representatives, and non-governmental bodies’ representatives) were purposively invited via email. They were further offered to participate in subsequent in-depth interviews in Stage 3.

#### 2.1.3. Data Analysis

Responses for open ended questions were read and checked for completeness. The responses were exported to NVIVO version 12 for coding and content analysis.

Research team members performed content analysis independently by coding the responses, followed by series of group discussions to reach consensus. The initial coding was to explore issues pertaining to Malaysia’s health systems response towards COVID-19 by the 8 pillars of health systems’ preparedness and response areas.

### 2.2. Stage 2: Mapping Issues Based on Health Systems Building Blocks

Issues identified from stage 1 were further mapped according to the health systems building blocks [1]. In-depth interviews (IDI) were conducted subsequently among experts to explore additional issues and recommendations for HSS during an outbreak. This step was adopted as the pillars focused on the steps or processes needed to manage the outbreak efficiently, while the aim of the study was to identify gaps in health systems’ strengthening. Identification of the gaps through building blocks or core structures of health systems shifted the focus to long-term efforts towards specific areas of the health systems, rather than on the processes which may change or influenced by various factors over time.

#### 2.2.1. Data Mapping

Further regrouping of issues into core health systems’ areas was guided by WHO Health Systems building blocks, namely: (1) leadership and governance; (2) health information; (3) health financing; (4) health workforce; (5) medical products; and (6) service delivery. This mapping process was done through iterative group discussions, discussing discrepancies among team members until consensus was achieved.

#### 2.2.2. In-Depth Interviews (IDI)

We conducted a virtual interview (Zoom video conference) among experts who expressed interest to participate in the IDI session. The experts confirmed their attendance through online invitations, and participated in an hour-long session, moderated by the research team members upon attaining their consent verbally. The interviews were audio recorded via the platform with permission from the participants. The IDI enabled further explorations and understanding on health systems issues in responding towards COVID-19 with further probing on suggestions for HSS within the 6 areas of the WHO Health Systems Building Blocks.

#### 2.2.3. Data Analysis

The interviews recordings were transcribed, checked for accuracy and validated by the researchers trained in qualitative method. The transcripts were imported to NVIVO version 12, QSR International Pty Ltd. (2018) for coding and content analysis. The coding process was done in the group, with discussion and consensus from the team members. Data were coded into issues and areas for improvement.

### 2.3. Stage 3: Identifying Gaps in Health Systems’ Strengthening

The next step involved consolidation of issues identified of each building block into health systems’ interactions. This was based on the acknowledgement that health systems were multifaceted and the blocks influenced one another, such that the interactions should be emphasised for strengthening strategies. This was achieved through discussions and resolving discrepancies among team members.

## 3. Results

### 3.1. Socio Demographic of Respondents

A total of 76 respondents completed the survey with their sociodemographic characteristics presented in Table 1. The respondents represented various backgrounds with different work experiences such as health and non-health related, providing us with rich information. Among the respondents, 3 public health specialists were involved in the IDI.

### 3.2. Issues of Health Systems’ Response towards COVID-19

In view of the complexity of the COVID-19 management requiring urgent actions to be taken to prevent further case escalation, most planning and executions of disease containment activities had to be promptly decided by the authorities. At the time of data collection, the COVID-19 outbreak was still at the early phase in Malaysia where the respondents highlighted multiple issues noted during this period of time.

Various issues were reported, and grouped into multiple domains and sub-domains in each pillar. For example, some suggested assessment of capabilities and significant contribution from various parties other than the main ministries including private agencies, industries and non-governmental bodies should be done for a more comprehensive strategic planning and decision-making in Pillar 1. Correspondingly, issues were also raised on the extent of collaboration, for example, the involvement of subject matter experts in assessing, analysing and generating input in outbreak management.

The health system must have a well-prepared multi sectoral national crisis planning developed based on an extensive risk assessment to formulate effective strategic plans. Any implementation or regulation must therefore be evidence-based and the legal basis considered. Respondents argued that with such approach, a timelier activation of strategies could have been in place. They also highlighted the need for an avenue to provide feedbacks to the governing body as well as an efficient platform to help implementers in making crucial decisions while facing the unprecedented but rapidly evolving COVID-19.

Respondents also highlighted the issues pertaining to supporting the surveillance activities. For example, respondents argued on the availability of standardised guidelines for data collection. Likewise, respondents also reported the absence of clear guidelines and regulations for adoption of correct infection and prevention control measures. Other issues were summarised in Table 2.

### 3.3. Health Systems’ Response in Outbreak Management

Issues identified from the pillars were mapped to the six areas of health systems building blocks for further understanding on the response towards the outbreak management (Table 3). The issues and areas for improvements identified pertaining to the steps taken in managing COVID-19 were converted into focused health systems’ areas.

### 3.4. Complex Interactions in Health Systems Strengthening

While the issues were essential within each building block, the interactions were fundamental for HSS. Figure 2 illustrates these complex interactions where building blocks influence each other (i), the leadership and governance providing the basis to all building blocks (ii), and service delivery block as an output to the interactions (iii).

#### 3.4.1. Interplays between the Building Blocks

There were many interactions seen between the building blocks. Decision-making processes in an outbreak relied heavily on the availability of timely and precise data. Utilisation of technology in data collection as well as guidance from subject matter experts within and outside the MOH were prerequisites for accurate data generation. An example highlighting this was the decision to lockdown the country in Malaysia [27]. Many expressed the usage of technology such as geographical information systems (GIS) and personal data for cases tracking would facilitate and allow risk stratification by zone, where lower risk areas could have less stringent movement restrictions, balancing between lives and livelihoods.


*“If this is being properly done with technology that every single case and their movement being mapped out during the early stage, I would think nationwide MCO is not really necessary,”*
(public, survey).

A country’s health financing status and model have huge impacts during a crisis. Not only a country’s reserve was crucial in managing and curbing the outbreak, it also determined the ability to sustain essential services and recuperate from the economic implications that came along with the outbreak [34]. Procurement and distribution of essential commodities, provision of technology, and adequate treatment measures were deemed necessary to manage the outbreak regardless of the country’s financial status. Likewise, allocation was also needed to train the health workforce to be competent in performing surveillance as well as compliance with infection and prevention control.

#### 3.4.2. Leadership and Governance Providing the Basis

Leadership and governance are commonly recognised as the key drivers influencing the functionalities of all building blocks, mostly due to the health systems dependency on decisions made by the stakeholders in responding to a crisis [35,36,37]. This includes having strategic vision among leaders at all level and effective coordination towards achieving a common goal, within and beyond the health systems. Strategic and timely coordination was the catalyst for resource optimisation and workload distribution. The expansion of laboratory testing capacity and enhancement in technology utilisation through the collaborative efforts between government agencies, universities and industries would enable fast disease detection and containment. With a centralised administrative system led by the MOH, COVID-19 responses such as strategic mobilisation of health workforce and standardised operating procedures implementation at all levels could be well-coordinated and synchronised effectively. However, such centralised command line could potentially restrict the local governments in making timely decisions for the best of their communities.


*“You are not dealing with it in a holistic manner, when (central governance) may not understand the demographics of the state, the dynamics of the population and so on... and tried to apply something which may not be resonant with the (state) ordinance itself. The power of the state to deal with immigration for example is up to the state... whether they want to allow or not people coming in from outside, as the public health ordinance is always peculiar and different... I personally think the (state) health department should be given more leeway so that they can work better,”*
(Public Health Specialist, IDI).

Active participation of key leaders/players and the transparency of handling the issues could facilitate powerful systems strengthening effect and building trust among the community [2,32]. Issues on transparency of information shared with the public were raised, as widely available data would ensure information conveyed was evidence-based and reliable. Likewise, governing bodies’ transparent processes and actions were also deemed crucial to gain public trust.

Apart from gaining public trust, the implementors/front liners felt that outbreak related policies and preparedness should be frequently reviewed and standardised at federal (central) and regional for uniform responses. Policy addressing data sharing between agencies are also highly welcomed by local experts as this encourages their participation and contributions in fighting against the outbreak.


*“MOH must learn to trust and work with others. We have enough regulatory and administrative mechanism to ensure control of confidentiality and ownership of data if that is the main concern. One example is the signing of non-disclosure agreement (NDA),”*
(public, survey).

Leaders must consider prioritising strengthening efforts in outbreak management. Certain elements needed immediate attention as the outbreak unfolds, while others required proper deliberations and long-term development processes. For example, establishing clear guidelines and case definition, mobilising resources to areas in need, and accurate data analysis needed to be addressed immediately as these affect the outbreak handling, while policy improvement, competency development, and finance remodelling called for continuous strengthening efforts over a longer period and beyond the crisis.


*“Missing link between laboratory and epidemiology. Increased demand for testing was anticipated but additional incoming budget is too late and too little,”*
(public, survey).


*“The government has to think about building financial reserves to support events like this. Past reserves, meaning reserves which are separated from your foreign exchange reserve and separate from your other things,”*
(Public Health Specialist, IDI).

#### 3.4.3. Service Delivery as the Outputs

Efforts described by far eventually affected the quality and outreach of services delivered during the outbreak. Strong decision-makings, stable finance, adequate resource allocations as well as trained health workforce were required not only to curb the spread of the outbreak, but also to maintain the core health functions and ensure management of other illnesses were not jeopardised. For example, during the outbreak efforts were made by the government to accommodate COVID-19 cases such as preparing isolation wards and reinforcement of ICU while attending to the needs of patients with other diseases. This is impossible without clear policy and preparedness plan. Diagnostic testing was another example raised by the respondents where efficiency could be improved, reducing the turnaround time by collaborating with other agencies. However, standardisation of testing protocol is crucial to ensure quality and reliability of the results including national policy to monitor of sampling and laboratory testing process across agencies.


*“During big outbreak, MOH should engage university lab or private lab. To do this, we need all those potential laboratories to be designated,”*
(public, survey).


*“...Need to monitor how samples are obtained by private labs. If both oropharyngeal and nasopharyngeal swabs are needed, there is a need to monitor if private labs are testing samples obtained from both or only one,”*
(public, survey).

During the early stage, there was also concern with lack of service outreach to vulnerable population. This unsettled issue might be due to tracking abilities of MOH to access them, as well as lack of regulation on public activities or mass gathering and movement of foreign workers. Established guidelines and cooperation with other agencies and local communities were crucial to ensure population receive needed health services.


*“Most vulnerable population (i) hidden illegal immigrants, (ii) foreign worker crowding (iii) nursing home population, still not addressed,”*
(public, survey).

## 4. Discussion

As health systems are multifaceted, complex interactions were seen influencing the process management in dealing with an outbreak like COVID-19. These complex interactions included interplays among the building blocks, cross cutting effect of the leadership and governance building block, and service delivery building block as an output of the interactions. In order to strengthen the health systems and prepare for future calamity, efforts to address these interactions is imperative.

The nature of COVID-19 disease by itself complicated the management plan, even before consideration of other factors. The R0 of COVID-19 as estimated by the WHO was 1.4–2.5, indicating high transmissibility and self-sustaining of the disease unless controlled by effective measures [38]. With the pandemic COVID-19 continues to challenge health systems all over the world, it provided an avenue to reflect upon and redesign the systems. Past experience revealed failure to contain an outbreak was related to the weaknesses of the health systems, leading to massive social disruption and collapse of services [39,40].

*Addressing interplay between building blocks*. Health systems consisted of interlinked components interacting within the context in which the systems lied in [41]. These interplays among elements within the health systems formed networks of feedback cycles, often with unpredictable and non-linear linkages between the cause and effect of an implementation, generating a ‘dynamic complexity’ [42]. In many ways, this study revealed these complexities, whereby every remote action taken in one building block had repercussions in almost all other building blocks. Using the MCO as an example, preparedness and implementations of the orders were seen as to be needing improvement, as it had huge and pervasive socioeconomic implications [43]. The decision must be based on accurate existing data and reliable projection, while the implementation required collaboration with expert and utilisation of technology to guide a more structured, risk stratified execution. One example would be to properly delineate the allowable activities for low risk and higher risk areas, where the former could be exempted from the MCO, thus minimising the overall negative socioeconomic consequences. This realisation of the need to address multiple factors across many building blocks to make a single implementation successful was seen in many studies on health systems response towards an outbreak [11,44,45]. Particularly, a study found that in order to increase community access to health services, shortage of qualified workforce, poor workforce’ attitudes, poor relationship between workforce and the community, as well as waiting time and confidentiality protection confidentiality protection must be properly and adequately addressed [11]. The emerging idea was, instead of addressing an isolated issue, the more pressing need was to acknowledge the interplays within the elements and to formulate an overarching approach addressing all factors influencing the issue of interest.

*Importance of governance and leadership.* Apart from addressing the complex interlinkages between the building blocks, good leadership and governance ensure all approach are well thought of and implemented strategically. As interplays were context dependent, constantly changing, and relied on responses towards the measures taken [6], the adaptation ability of leaders and the governing systems towards contextual realities were deemed core components of successful implementations [16]. In countries adopting centralised health systems approach such as Malaysia, the policies and programmes were centrally formulated, funded and administered [41]. While such hierarchical administrative system has shown great resilience in dealing with outside forces [42], it left little room for feedbacks and influence from external parties. Various leadership approaches have demonstrated different outcomes in dealing with COVID-19. Each approach had its own strengths and weaknesses [46]. Regardless of the approach, it was argued that in an outbreak of such magnitude, governance should be approached with flexible and accommodative manner, including transforming the leadership and systems design during the crisis for a better adaptation [2]. The need for good governance with strategic visions at all level was therefore pivotal to ensure success. This concern was in line with many studies done identifying crucial roles of leaders in determining the outcomes during a disaster or programme implementations [41,44,45,46].

*Impact on service delivery.* Understanding of interconnectedness and complexity prior to the development of policy and actions plan should be emphasised as this would affect the service deliveries [45]. An example of a service delivery affected was the unmet capacity of diagnostic testing. While the numbers of tests conducted were increasing throughout the outbreak, earlier identification, engagement and collaboration with universities and private sectors were the areas reported requiring improvements for the country to better cope with the surge of cases. This finding was comparable with studies done in many countries showing how the complex interactions interfered with the outcomes [9,10,11,44,45]. In a study looking at Zika’s outbreak management, it was found strong collaboration among stakeholders and efficient surveillance systems generating database of Zika-positive pregnant women shared effectively among healthcare providers were the main drivers for successful service deliveries [9]. In a nutshell, health systems relied on smooth and efficient interactions between the fundamental elements to deliver quality services while facing a crisis [47].

*Implications for health system strengthening.* Perhaps, a constantly proposed solution in addressing the complexity of health systems’ interactions is to adopt systems thinking approach. It provides structured solutions to address complex issues especially in designing the outbreak management, assessing the gaps and areas for improvements, as well as planning implementations [6,11]. The systems must be viewed as a whole rather than its individual components, taking into account evolution of the systems’ behaviour and the constantly changing and dynamic roles of actors, key-players, as well as the population [2,42]. This would require establishment of clear policies and guidelines to achieve seamless and transparent collaboration between various sectors, ministries and even departments [46]. This also means all relevant parties including community, private sectors and industries must be engaged to generate the best approach to manage issues at hand [44]. This must be followed by effective communication to channel information to the public. Finally, addressing complexity and health systems’ interactions also entail capacity to differentiate and prioritise issues to be addressed as strengthening efforts need proper time allocation to develop sustainable interventions [44]. Some areas of improvements required immediate or urgent attention as the outbreak unfolded, while other areas required proper deliberations beyond the outbreak. This droves well with the discussion on differentiating between health systems support and strengthening, whereby strengthening efforts require comprehensive and permanent changes to the systems design [3].

*Strength and limitations.* When health systems are resilient, the population is protected. The strengthening efforts as suggested by the study will catapult the attainment of a resilient health system in dealing with crisis, of which the system continuously makes improvements based on lessons learned, and develops ability to withstand and adapts to any crisis among all its elements [47]. Analysing the findings based on the health systems building blocks offered myriads of advantages. Apart from being commonly used as the language to describe and compare health systems, the building blocks incorporated an overarching, holistic health systems viewpoint by allowing a better view of each health systems’ area as well as the interactions and linkages between the areas [1,11,48]. While the improvement of processes involved in outbreak management were in reality more practical suggestions, the building blocks allowed better understanding of the long-term strengthening efforts required as the areas were less robust to changes in comparison to the processes. The building blocks merged the concept with the operationalisation, allowing strengthening of both the areas and the processes, as well as reducing duplication of efforts and resources. The exercise also allowed exploration of areas not well covered by the instrument used in the study. We realised health financing was not given much emphasis in the SPRP document, and therefore, was further explored during the in-depth interviews. Such exercise was previously done by other study [49]. As the study explored public’s perspective, all responses were subjected to their opinion and experiences which may not necessarily reflect the reality. We suggest further triangulation with other sources of information including review on all strategies taken by the country to increase the comprehensiveness of the findings. The study was conducted at the early phase of COVID-19 in Malaysia, whereby many efforts were just formed and implemented. Many subsequent strategies were adopted, which may have addressed issues reported by respondents of this study. Nevertheless, the feedbacks were still relevant for future strengthening efforts.

We used multiple platforms to reach as many and diverse respondents, however as the survey utilised was based on the WHO SPRP documents, the questions were complex and specific, rendering only those who could understand health systems management of COVID-19 able to answer the survey. Future study exploring public perception on COVID-19 health management should target wider and more representative respondents with a more simplified tool. Using open-ended questions survey enabled respondents to express their opinions freely without restrictions. Researchers commonly faced challenges whereby the result may be brief or having no response as open-ended questions require more time and effort. However, this limitation was overcome in our study by followed-up interviews which provided respondents the opportunities to further elaborate on their thoughts. Another limitation was restricted applicability of the findings to other crises beyond the specified outbreak. Since the study gathered opinions specific to the COVID-19 context, many issues were bounded by the circumstances around the disease, which may be different or not applicable to other crisis situations. Further study exploring HSS in facing all crisis is the next step forward. Finally, we also recommend further exploration among the stakeholders directly involved in the outbreak management on the success and failures of implementations as well as the transformation needed for better crisis management.

## 5. Conclusions

The judicious approach to understanding the propellers of health systems strengthening proposed in this study has potential to help policymakers to understand the complexities in fostering whole systems change through a holistic measure in facing an outbreak. Specifically, the study demonstrated the complex interlinkages between the building blocks, cross-cutting effects of leadership and governance, and service deliveries affected by all other building blocks, indicating the need to address issues identified at a macro level. We suggest application of systems thinking as a pragmatic solution, addressing each identified issue including considering how an action will implicate other building blocks and approaching the governance effort in a more flexible and adaptive manner. Despite the identified issues being context specific to Malaysia, mapping them to the building blocks highlighted the complexity of health systems strengthening at a more abstract level, thus applicable to all health systems with varying applications based on country specific context. This entails development of country specific action plan formulating short and long-term strategies addressing each identified issue. This would propel towards development of resilience health systems where they are able to prepare for, adapt, transform, and continuously learn from any crisis. When health systems are resilient, the population may well be protected.

## Figures and Tables

**Figure 1 ijerph-18-09047-f001:**
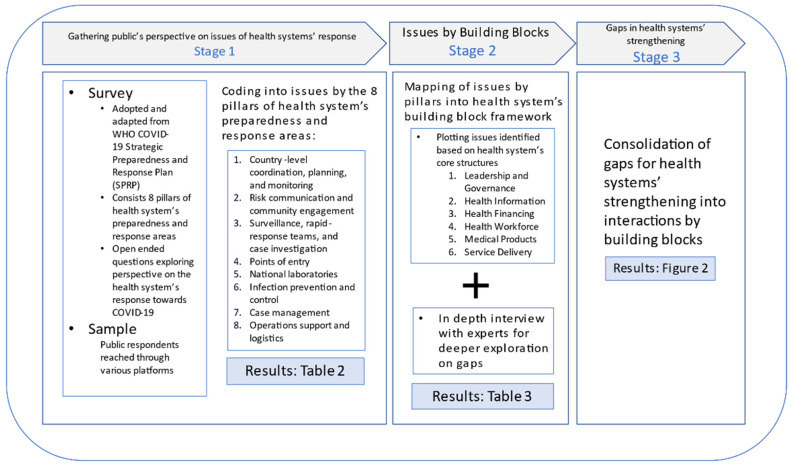
Summary of study flow based on the three stages.

**Figure 2 ijerph-18-09047-f002:**
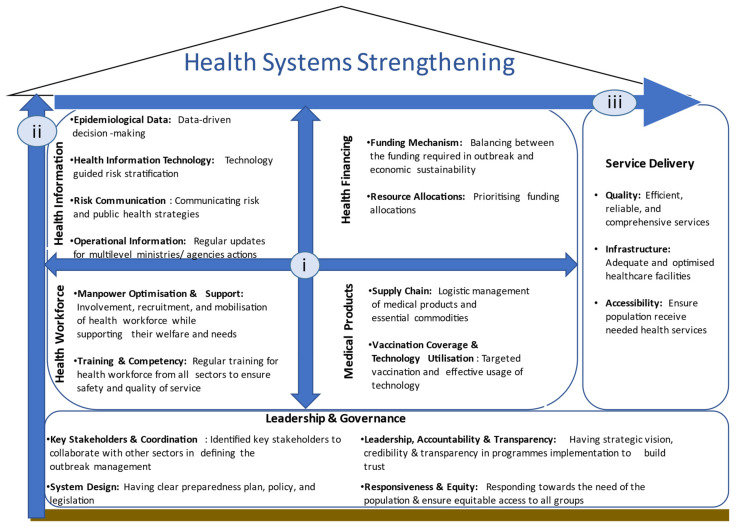
Complex interactions of health systems strengthening.

**Table 1 ijerph-18-09047-t001:** Respondents’ characteristics (*n* = 76).

Characteristics	Respondents, *n* (%)
Total	*n* = 76
Affiliation	
Government	29 (38.1)
Private	20 (26.3)
NGO/Association	4 (5.2)
No affiliation	9 (11.8)
Not disclosed	14 (18.4)
Professional Background	
Health related	55 (72.3) *
Non health related	7 (11.7)
Not disclosed	14 (23.3)

Notes: * three public health specialists involved in the IDI session.

**Table 2 ijerph-18-09047-t002:** Perception on health systems response at the early phase of COVID-19.

Pillar 1: Country Level Coordination, Planning and Monitoring		
Domains	Sub-Domains	Issues
Stakeholder	Stakeholders’ identification	1. Lack of assessment of capabilities/availabilities of all stakeholders including private agencies, industries, non-governmental organisations, experts, and community representatives
		2. Late involvement of key stakeholders
	Stakeholders’ cooperation	1. Lack of coordination across ministries and industries
		2. Lack of coordination between central and regional authorities
		3. Lack of involvement of experts for assessment, analysis and input generation for outbreak management
Process Implementation	Managerial approach	1. Lack of preparedness plan, policy and guideline
		2. Lack of evidence-based statement in information dissemination and action
		3. Timeliness of planning and activation
	Synchronised and coordinated instructions	Contradicting implementations between federal and regional authorities
	Monitoring, evaluation and documentation	1. Comprehensiveness and credibility in evaluation of policy/action/performance measures
		2. Need for lesson learned documentation
	Transparent planning, implementation and performance measure	Lack of transparency of action plan, assessment and performance measure towards public
**Pillar 2: Risk Communication and Community Engagement**		
**Domains**	**Sub-Domains**	**Issues**
Dissemination of Information	Source of information	Lack of credibility in information dissemination to ensure information disseminated is accurate and evidence-based
	Content of information	1. Unclear information due to technical terms, language and incomplete information
		2. Inconsistent information
		3. Information disseminated is not tailored with community’s level of acceptance
	Relay of information	Inadequate channels and platforms to reach all population
Community Involvement	Extent of involvement of relevant parties	1. Lack of coordination with community representatives
		2. Untimely engagement
	Receptiveness on severity of outbreak	Not preparing the community prior to instruction dissemination
**Pillar 3: Surveillance, Rapid Response Team and Case Investigation**		
**Domains**	**Sub-Domains**	**Issues**
Surveillance Structure	Legislation on data sharing	Incomprehensive legislation on sharing of surveillance data
	Extent of network and collaboration structure	1. Lack of networking and partnership with experts on data analysis
		2. Lack of coordination in data integration
Surveillance Functions	Case definition and guideline	1.Lack of clear case definition and strong team to detect all including isolated cases
		2. Lack of standardised guidelines
	Data collection and handling	1. Inaccurate data analysis and interpretation
		2. Inefficient centralised platform to communicate and disseminate data
		3. Inadequate training to improve competency
		4. Unoptimised resources for efficient tracing, data collection and data analysis
Surveillance Quality	Data quality	1. Lack of completeness and transparency
		2. Lack of representativeness and accuracy
**Pillar 4: Point of Entry**		
**Domains**	**Sub-Domains**	**Issues**
Entry/Exit Control	Travel restriction	Lack of stringency in travel restrictions and country entry requirements
	Border closure	Untimely border closure
Screening measures	Stringency	Lack of stringency in screening measures and regulations at point of entries
	Competency	Incompetent health workers implementing and explaining needs for screening measure
	Consistency	Inconsistent method of screening measures at all entries including air, land and sea
Quarantine policy	Policy implementation	1. Untimely policy implementation
		2. Unclear instructions and advice for people instructed for quarantine upon entering the country
		3. Lack of tracking and follow up measures of people under quarantine
Information	Update on disease management and standard operating procedure (SOP)	Untimely and inefficient relay of information and update on SOP due to rapidly changing situation
**Pillar 5: National Laboratories**		
**Domains**	**Sub-Domains**	**Issues**
Partnership and Networking	Co-ordination with universities, private laboratories and GPs	1. Lack of national level coordination with university and private laboratory for testing and research
		2. Inadequate information for management at general practitioner (GP) level
		3. Lack of collaboration in test kit development to increase diagnostic capacity
Laboratory Testing	Test criteria	Rigid test criteria affecting testing coverage
	Volume of testing	Inadequate testing for suspected case
	Sample transportation	Difficulty of transporting samples from East Malaysia
	Testing efficiency	Inefficient testing process causing delay of result
Surge phase preparation	Anticipation of surge phase	Lack of preparation leading to inability to cope with surge of cases
Financing	Cost and budget availability	1. More out of pocket money to do test in private sector
		2. Shortage of reagents and other inventories due to lack of fund
Monitoring and Evaluation	Monitoring screening process	Questionable quality of sample taken in private hospitals and clinics
Human Resources	Health workers mobilisation	Lack of and untimely mobilisation of human resources to busy centre
	Training	1. Inadequate training especially to private healthcare workers
		2. Need for GP empowerment
Procurement and Inventory Management	Supply	Inadequate supply of reagents/swab/personal protective equipment to cater for increase number of testings
	Evaluation of test method	Lack of evaluation on selection and usage of rapid testing
**Pillar 6: Infection Prevention and Control**		
**Domains**	**Sub-Domains**	**Issues**
IPC programmes and policies	Facility lead team	Lack of dedicated and trained lead team to ensure infection prevention and control (IPC) programme at facilities run as planned
	Clear directives and policy	1.Unlear directives to help healthcare staffs adhere to IPC protocol
		2. Incomprehensive policy on public activities: mass gathering/punitive action/mask wearing/religious places
		3. Incomprehensive policy for workplace: shifts/numbers/protocols/SOP
		4.Incomprehensive policy on domestic travelling
	Enforcement	Weak enforcement of IPC protocol
	Risk stratification	Not utilising technology to risk stratify area and activities
	IPC training	1. Lack of allocation of time and budget for healthcare workers’ training
		2. Lack of empowerment of non-healthcare workers in healthcare facilities
	Public awareness creation	Need to increase awareness on basic measures such as proper handwash and wearing mask
Healthcare workers surveillance	Testing HCW testing	Healthcare workers should be given priority in doing testing
Monitoring and documentation	Monitoring and Analysis	Lack of monitoring and analysis of IPC programme. For example; healthcare workers adherence to IPC Protocol
	Documentation	Need for proper documentation of cases and lesson learnt
Built environment, materials and equipment	Infrastructure and engineering approach	Lack of proper isolation facility with good ventilation to quarantine positive cases
	Resources and stockpiling	Lack of resources and stockpiling
	Quarantine centre management	Inefficient organisation of quarantine centre
	Infrastructure and services	Inappropriate infrastructure and services to encourage public to adhere to IPC
**Pillar 7: Case Management**		
**Domains**	**Sub-Domains**	**Issues**
Staff Training	Training and involvement of primary care providers	Lack of training and involvement of primary care providers in case management
Facilities	Capacity of treatment and quarantine centres	1. Inadequate intensive care units and tertiary care capacities
		2. Lack of quarantine centres
Treatment	Medical advancement	Incomprehensive utilisation of other COVID-19 treatment (example: plasma collection of convalescing COVID-19 patients)
Information and communication	Guidelines and information dissemination to public	Not using targeted approach (communication) in disseminating information
	Guidance and protocols for healthcare providers	1. Lack of guideline in managing patient transferred from MOH to private hospitals
		2. Unclear guidance on how to manage patient with mild symptoms
		3. Lack of single (centralised, key point) information source for both public and private healthcare workers
		4. Lack of guidance for general practitioner and primary care centres in patient screening and management
		5. Lack of protocol for private sector to support public health operationalisation
**Pillar 8: Operational Support and Logistics**		
**Domains**	**Sub-Domains**	**Issues**
Supply chain	Landscape assessment	1. Inadequate review of emergency supply chain process
		2. Lack of preparedness in operational plan
		3. Party monopoly in supply provision
	Governance, financing and personnel	1.Lack of coordination on supply chain processes
		2. Unclear directive (hospital on their own)
		3. Lack of support to obtain equipment and set up
		4. Unclear fund release process
	Emergency protocol	Lack of protocol for emergency resources mobilisation
	Commodity planning and quantity forecasting	1. Not addressing shortage of PPE among general practitioners
		2. Poor resource forecasting
	Procurement and sourcing of emergency health commodities	1. Poor control on exports of materials
		2. Lack of planning and procurement process
		3. Not prioritising and supporting local suppliers
		4. Lack of centralised procurement effort
	Stockpiling and Warehousing	Not identifying reserve for basic field equipment
	Distribution of supply	1. Lack of coordination in distribution
		2. Lack of transparency in distribution
	Logistics Management Information Systems	No feedback of transparency
Human resources	Capacity	1. Need for more recruitment of contract and volunteer workers
		2. Inadequate identification of human resource requirement and deployment of resources
	Welfare	1. Need for supplementary vaccination
		2. Inadequate social and emotional support

**Table 3 ijerph-18-09047-t003:** Issues of health systems response according to the building blocks.

Building Block: Leadership and Governance	
Domains	Issues/Areas for Improvements
Key stakeholders	1. Challenge in identifying key stakeholders in designing the national action plan and policy
	2. The need for a multi-sectorial council with involvement of private agencies, industries, non-governmental organisations, experts of various backgrounds, community representatives
	3. Timely involvement of key stakeholders
Collaboration and coordination	1. Expansion of coordination, engagement and input generation across ministries and industries
	2. The need for multisectoral simulations
Systems design	1. Call for development of a more comprehensive and adaptive national preparedness plan as many decisions in current pandemic management were seen made on impromptu basis
	2. Some existing national policies and past implementations are not reviewed and pursued in current outbreak
	3. Need for clear protocols and guidelines beyond case management and prevention control
	4. Lack of standardisation on coordination of instructions and implementations across ministries as well as between federal and regional authorities within the ministry
	5. Confusing, frequently changing instructions and lack of monitoring leading to breach of protocols among staff in both public and private healthcare facilities
	6. Effectiveness of centralised approach in managing the outbreak is debatable
Leadership	Leaders must have strategic vision
Accountability	Credibility and trustworthiness—call for evidence-based dissemination of information and enforcement, and addressing data ownership and confidentiality as means to extend collaboration with experts from other sectors
Transparency	Lack of transparency in planning, implementation and performance measures
Responsiveness	Call for the government to attend urgently to the following needs:
	1. Obtaining essential items in handling the outbreak
	2. Limited infrastructures to handle COVID-19 in many healthcare facilities
	3. Lack of human resources to manage COVID-19
Equity	Attending to the health needs of the marginalised and vulnerable groups during the outbreak
**Building Block: Health Information**	
**Domains**	**Issues/Areas for Improvements**
Epidemiological data	1. Inaccurate epidemiological data in informing decision makers of public health intervention
	2. Lack of adherence to epidemiological principle in surveillance data, analysis and interpretation determines the quality of information
	3. Lack of data sharing among the experts (e.g., epidemiologist, public health experts) leads to unoptimised expertise utilisation
Health information technology	Need for advanced health information technology for disease surveillance in contact tracing, enforcement and disease modelling activities to automate data generation with higher accuracy
Risk communication	1. Accuracy of information provided by the authorities on the outbreak status and progression is debatable
	2. Lack of channel/platform for information dissemination which should be timely, accessible, sufficient, consistent, and transparent
Operational information	Need for clear and updated information to support the health workforce and frontliners at the ground to implement effective interventions
**Building Block: Health Financing**	
**Domains**	**Issues/Areas for Improvements**
Funding mechanisms	Current financial resources for outbreak management depended on ad hoc funding which might not be sustainable for future crisis
Resource allocations	1. Urgent allocations were needed for various infection control activities, putting more strain to the healthcare budget
	2. Prioritisation of government budget for outbreak management was needed to support efficient implementation
**Building Block: Health Workforce**	
**Domains**	**Issues/Areas for Improvements**
Manpower optimisation and support	1. More effort needed to recruit volunteers and retired staff, as well as mobilising workforce from less affected areas to epicentres
	2. Need to strengthen the primary care and public health to ease the burden of surveillance and contact tracing
	3. Lack of adequate support and appropriate incentives to healthcare workers
	4. Need for whole-of-country approach with a task force consisting of subject matter experts from various fields with the required knowledge, skills, and expertise from within and outside of the government agencies
	5. Community representatives inadequately involved with lack of coordination and communication between the government agencies and the representatives
Training and competency	No regular training in outbreak management for diverse group of healthcare workers performing critical functions often due to the lack of resources
**Building Block: Medical Products, Vaccines and Technology**	
**Domains**	**Issues/Areas for Improvements**
Supply chain	Ensuring adequacy and access to medical products through proper planning and forecasting with consideration of both the public and private healthcare facilities
Vaccination coverage and technology utilisation	1. Investing in the required infrastructure to develop the necessary technology locally
	2. Prioritising high-risk groups to be vaccinated, along with economic evaluation
**Building Block: Service Delivery**	
**Domains**	**Issues/Areas for Improvements**
Quality	1. Health systems must be responsive to outbreak without compromising the core functions and service delivery quality
	2. Need for quality service delivery during the outbreak: efficient, reliable and comprehensive
Infrastructure	Urgent need to optimise and upscale existing infrastructures and swiftly open up temporary units such as quarantine centres
Accessibility	Lack of service outreach to vulnerable population

## Data Availability

The dataset that supports the findings of this article belongs to the Malaysia Health Systems Response in COVID-19 study. At present, the data are not publicly available but can be obtained from the corresponding author and Head of Centre for Biostatistics & Data Repository, National Institutes of Health, Ministry of Health Malaysia on reasonable request and with the permission from the Director General of Health, Malaysia.

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
