# Peer review of "Addressing Gaps for Health Systems Strengthening: A Public Perspective on Health Systems’ Response towards COVID-19"

_ijerph, 2021, doi:10.3390/ijerph18179047_

Round 1

Reviewer 1 Report

it is a good paper and with only one minor comment I checked the "minor revision". My only one minor comment was about the implications in the conclusion to be more attractive for the international readership.

The paper is concerned with a current important topic with interesting insightful findings. The process is well described and the figure of the study stages helps the reader understand. It is advised, in order to make the paper more interesting to the international readership, to provide more concrete implications in the conclusion section and more specifically, implications that can be useful internationally. 

Author Response

Response:

Thank you for your feedback. We have expanded the conclusion to explain how the results are applicable to all health systems despite being conducted in Malaysia. We also added proposal to develop country specific action plan to address all issues identified in managing COVID-19.

Changes:

Line 538 – 544 in tracked changes:

Despite the identified issues being context specific to Malaysia, mapping them to the building blocks highlighted the complexity of health systems strengthening at a more abstract level, thus applicable to all health systems with varying applications based on country specific context. This entails development of country specific action plan formulating short and long-term strategies addressing each identified issue. This would propel towards development of resilience health systems where they are able to prepare for, adapt, transform, and continuously learn from any crisis.

Reviewer 2 Report

The article presents very important issues concerning the functioning of health systems in most societies affected by the SARS-Cov-2 virus pandemic. The strength of the article is a detailed description of the research methodology, methods of data archiving and methods of the authors' efforts to objectify the collected data. The latter issue may raise doubts among readers due to the lack of representativeness of the adopted sample of respondents and the reference to selected statements of experts who were not indicated by name and surname.

In my opinion, the title of the study should be modified and inform about the situation identified in Malaysia. The authors of the study indicate that the course of events in other countries did not coincide with the situation described in the article. I also propose to present the quoted statements briefly and in tabular form.

Author Response

Comment 1)

The article presents very important issues concerning the functioning of health systems in most societies affected by the SARS-Cov-2 virus pandemic. The strength of the article is a detailed description of the research methodology, methods of data archiving and methods of the authors' efforts to objectify the collected data. The latter issue may raise doubts among readers due to the lack of representativeness of the adopted sample of respondents and the reference to selected statements of experts who were not indicated by name and surname.

Response:

Thank you for your feedback and we agree with your concern on lack of representativeness of respondents in this study. However, as the questions posed in the survey utilized in this study are complex and focused on health system, we had limited number of responses despite advertising in various platforms. It also explains the narrowed background of our respondents, whereby only those who understand health system management of COVID-19 were able to answer the survey. For future study exploring public perception on COVID-19 health management targeting wider and more representative respondents, we would suggest to use a more simplified tool. This justification is now added into the limitation section under Discussion.

Changes:

Discussion; Line 510 – 515 in tracked changes

We used multiple platforms to reach as many and diverse respondents, however as the survey utilised was based on the WHO SPRP documents, the questions were complex and specific, rendering only those who could understand health systems management of COVID-19 able to answer the survey. Future study exploring public perception on COVID-19 health management should target wider and more representative respondents with a more simplified tool.

Comment 2)

In my opinion, the title of the study should be modified and inform about the situation identified in Malaysia. The authors of the study indicate that the course of events in other countries did not coincide with the situation described in the article.

Response:

Thank you for your feedback. Although COVID-19 challenges are context and country specific, and the findings are Malaysian based, we would like to leave the title to be more general and highlight the main focus which are: Health system strengthening, health system management of COVID-19, and public perception. Expanding the title to include Malaysia or Malaysian findings would deviate the focus. We have included Malaysia as one of the keywords in the article.

Comment 3)

I also propose to present the quoted statements briefly and in tabular form.

Response:

Thank you for your suggestion. We decided to leave the quoted statements embedded in the paragraph for a better flow to facilitate readers’ understanding on the context and meaning of the statements. 

Reviewer 3 Report

the authors discussed a complex and elaborate topic. The design of the study is well described, as is the presentation of the results. 
I suggest to implement the literature in the introduction, citing some examples of rapid reorganisation of the health system, especially in the first phase of the pandemic (lines 62-67) (e.g. Thomas S, Sagan A, Larkin J, Cylus J, Figueras J, Karanikolos M. Strengthening health systems resilience: Key concepts and strategies [Internet]. Copenhagen (Denmark): European Observatory on Health Systems and Policies; 2020. PMID: 32716618 ; Patel A, Jernigan DB; 2019-nCoV CDC Response Team. Initial Public Health Response and Interim Clinical Guidance for the 2019 Novel Coronavirus Outbreak - United States, December 31, 2019-February 4, 2020. MMWR Morb Mortal Wkly Rep. 2020 Feb 7;69(5):140-146. doi: 10.15585/mmwr.mm6905e1. Erratum in: MMWR Morb Mortal Wkly Rep. 2020 Feb 14;69(6):173. PMID: 32027631; PMCID: PMC7004396; Patrizi A, Bardazzi F, Filippi F, Abbenante D, Piraccini BM. The COVID-19 outbreak in Italy: Preventive and protective measures adopted by the Dermatology Unit of Bologna University Hospital. Dermatol Ther. 2020 Jul;33(4):e13469. doi: 10.1111/dth.13469. Epub 2020 Jun 18. PMID: 32347635; PMCID: PMC7261988; Marasca C, Ruggiero A, Annunziata MC, Fabbrocini G, Megna M. Face the COVID-19 emergency: measures applied in an Italian Dermatologic Clinic. J Eur Acad Dermatol Venereol. 2020 Jun;34(6):e249. doi: 10.1111/jdv.16476. PMID: 32294282; PMCID: PMC7262301)

Furthermore, I suggest streamlining the discussion and emphasising the main teaching points, as it is quite demanding.

Author Response

Comment 1

I suggest to implement the literature in the introduction, citing some examples of rapid reorganisation of the health system, especially in the first phase of the pandemic (lines 62-67) (e.g.

Thomas S, Sagan A, Larkin J, Cylus J, Figueras J, Karanikolos M. Strengthening health systems resilience: Key concepts and strategies [Internet]. Copenhagen (Denmark): European Observatory on Health Systems and Policies; 2020. PMID: 32716618 ;

Patel A, Jernigan DB; 2019-nCoV CDC Response Team. Initial Public Health Response and Interim Clinical Guidance for the 2019 Novel Coronavirus Outbreak - United States, December 31, 2019-February 4, 2020. MMWR Morb Mortal Wkly Rep. 2020 Feb 7;69(5):140-146. doi: 10.15585/mmwr.mm6905e1. Erratum in: MMWR Morb Mortal Wkly Rep. 2020 Feb 14;69(6):173. PMID: 32027631; PMCID: PMC7004396;

Patrizi A, Bardazzi F, Filippi F, Abbenante D, Piraccini BM. The COVID-19 outbreak in Italy: Preventive and protective measures adopted by the Dermatology Unit of Bologna University Hospital. Dermatol Ther. 2020 Jul;33(4):e13469. doi: 10.1111/dth.13469. Epub 2020 Jun 18. PMID: 32347635; PMCID: PMC7261988;

Marasca C, Ruggiero A, Annunziata MC, Fabbrocini G, Megna M. Face the COVID-19 emergency: measures applied in an Italian Dermatologic Clinic. J Eur Acad Dermatol Venereol. 2020 Jun;34(6):e249. doi: 10.1111/jdv.16476. PMID: 32294282; PMCID: PMC7262301)

Response:

Thank you for your suggestions. We have added all suggested citations in the Introduction, expanding the countries experiences to past and more recent ones on COVID-19. We have also used some of the suggested citations to support discussion points.

Changes:

Introduction; Line 66 – 73 in tracked changes

Past and recent outbreak experiences have demonstrated countries with strong and established health systems were able to adapt and produce good health outcomes in contrast with vulnerable health systems which struggled to respond effectively to adverse conditions. Countries like Taiwan and South Korea leveraged on their past experience of SARS and MERS outbreaks which exposed their nation’s limitations and prompted improvements to be made [23-25]. Countries rapidly formed new approach and guidelines in adapting to the challenges brought upon by COVID-19 [20-22].

Discussion; Line 428 – 431 in tracked changes

As interplays were context dependent, constantly changing, and relied on responses towards the measures taken [6], the adaptation ability of leaders and the governing systems towards contextual realities were deemed core components of successful implementations [16]

Comment 2

Furthermore, I suggest streamlining the discussion and emphasising the main teaching points, as it is quite demanding.

Response:

Thank you for your feedback. We have added brief introductory statement in the discussion paragraphs to guide readers on what a particular paragraph is explaining and the main learning point, to ease understanding of the discussion. We have also removed some sentences to shorten the discussion. We have also expanded the conclusion to have more learning points.

Changes:

Discussion:

Removed line 466 – 471 in tracked changes

Brief introductory statements added for each paragraph.

Conclusion:

Line 538 – 544 in tracked changes:

Despite the identified issues being context specific to Malaysia, mapping them to the building blocks highlighted the complexity of health systems strengthening at a more abstract level, thus applicable to all health systems with varying applications based on country specific context. This entails development of country specific action plan formulating short and long-term strategies addressing each identified issue. This would propel towards development of resilience health systems where they are able to prepare for, adapt, transform, and continuously learn from any crisis.

 Thank you